# Impact of oral diseases on oral health-related quality of life: A systematic review of studies conducted in Latin America and the Caribbean

**María T. Yactayo-Alburquerque**[1]*, **María L. Alen-Méndez**[1], **Diego Azañedo**[2], **Daniel Comandé**[3], **Akram Hernández-Vásquez**[4]

1 Universidad Científica del Sur, Lima, Peru, 2 Independent Researcher, Lima, Peru, 3 Instituto de Efectividad Clínica y Sanitaria (IECS), Buenos Aires, Argentina, 4 Universidad San Ignacio de Loyola, Vicerrectorado de Investigación, Centro de Excelencia en Investigaciones Económicas y Sociales, Lima, Peru

* marite160896@gmail.com

## Abstract

### Background

We performed a systematic review of studies conducted in Latin America and the Caribbean (LAC) to assess the impact of oral diseases on oral health-related quality of life (OHRQoL).

### Materials and methods

Searches were performed of the following PubMed, EMBASE, CINAHL, Scopus, and LILACS databases. Randomized clinical trials, quasi-experimental studies, cohort studies, case and control studies, and cross-sectional studies which included at least 100 participants evaluating the impact of oral diseases on OHRQoL were included. PROSPERO registry number: CRD42020156098.

### Results

After exclusion of duplicates, 3310 articles were identified, 40 of which were included in this review. 90% of the studies were conducted in Brazil. The most commonly used OHRQoL measuring instruments were CPQ 11–14 (n = 9), ECOHIS (n-8) and B-ECOHIS (n = 8). The study designs included 32 cross-sectional, 2 cohort and 6 case and control studies. Most of the studies were conducted in children (n = 25) and adolescents (n = 9). Most studies identified an impact on OHRQoL in children, adolescents and adults with oral diseases. Moreover, greater oral disease severity had a greater impact on OHRQoL.

### Conclusions

Most studies in LAC report a negative impact of diseases on OHRQoL. More longitudinal studies are required to confirm the results of these studies.

**Data Availability Statement:** All relevant data are within the paper and its Supporting Information files.

**Funding:** The authors received no specific funding for this work.

**Competing interests:** The authors have declared that no competing interests exist.

## Introduction

Oral health is recognized as an essential component of quality of life; however, historically more attention has been paid to its mainly local clinical consequences, regardless of the impact these can have on people's daily lives [1]. The World Health Organization (WHO) defines oral health as a disease- and disorder-free state that limits a person's ability to bite, chew, smile and speak, as well as their psychosocial status [2]. Thus, the presence of oral diseases can affect a person's growth and development, as well as their psychic, productive and social capacity.

Oral diseases are a highly prevalent group of pathologies in the world. The 2017 Global Disease Burden Study reported that 3.5 billion people suffered from oral disease in 2016, representing nearly 50% of the world's population [3]. In Latin America, a significantly higher prevalence and incidence of untreated cavities in permanent teeth, severe periodontal disease and total tooth loss compared to global averages in 2015 [4] has also been reported. This situation is a priority, as it generates direct and indirect expenditures that affect the limited budgets of health systems in the countries in this region. Thus, while direct expenses are related to costs resulting from oral treatments, indirect expenses are related, for example, to the loss of productivity and quality of life as a result of pain, infections, loss of school days and family life disturbances caused by oral diseases [5].

At the beginning of the 1980s the concept of "oral health related quality of life" (OHRQoL) was introduced, which consists of a multidimensional construct that reflects the degree of perception of comfort and satisfaction of a person in their daily life regarding their oral health [6]. The complexity of measuring this construct in the different population groups created the need to develop specific instruments for its measurement and evaluation worldwide in order to determine how it is affected by the presence of oral diseases. Thus, systematic reviews have now been published assessing the impact of oral diseases on OHRQoL; however, these are limited to the study of certain pathologies such as: tooth decay [7], dental trauma injuries (TDI) [8] and malocclusion [9] mainly in minors. Therefore there is an underrepresentation of this type of studies in other oral pathologies such as periodontal diseases, temporomandibular dysfunctions, pathologies of the salivary glands, cleft lip and palate, and edentulism, as well as in the population of adults and older adults. In addition, the representation of Latin American and Caribbean (LAC) countries in the above systematic reviews is extremely poor, making further studies evaluating the relationship of oral diseases and OHRQoL in the region necessary.

In short, knowing the effect of oral diseases on OHRQoL is critical to public health systems, research, and decision-making on strategies to be implemented in the prevention and promotion of oral health in different countries. In addition, it is of concern that oral health care is often the first service considered to be expendable by decision makers, due to its high costs and the perception of being a non-essential service especially in developing countries [10]. Therefore, the objective of this study was to conduct a systematic review of studies conducted in LAC to assess the impact of oral diseases on OHRQoL.

## Materials and methods

For the systematic review report, the Meta-analysis of Observational Studies in Epidemiology (MOOSE) guide and the declaration of reference items for publishing systematic reviews and meta-analysis (PRISMA) [11, 12] were followed. The systematic review protocol was registered with PROSPERO with reference number CRD42020156098.

### Eligibility criteria

The review included randomized clinical trials, quasi-experimental studies, cohort studies, case-control and cross-sectional studies with at least 100 participants assessing the impact of

oral diseases (dental cavities, malocclusion, TDI, periodontal diseases, temporomandibular dysfunction, salivary gland pathologies, cleft lip and palate, and edentulism) on OHRQoL in people of all ages in LAC published over the past 10 years. For duplicate data in more than one publication, the study with the largest sample was selected.

*In vitro* or animal studies, editorials, systematic reviews, letters to the editor, conference proceedings, case reports, studies not conducted in LAC were excluded. In addition, studies evaluating oral pathologies associated with systemic diseases or secondary diseases in the oral cavity were excluded.

## Sources of information

A systematic electronic search was conducted in the databases of PubMed, Embase, CINAHL, Scopus, and LILACS (Latin American and Caribbean Literature in Health Sciences) of articles published until July 18, 2020.

## Search strategies

A search strategy initially designed in PubMed was adapted to the other databases. A librarian (DC) developed the search strategies that were validated by two of the authors (AHV and DA). The search terms are detailed in S1 Table of the supporting information. The search was performed without restrictions of study design, publishing status, or publishing language.

## Selection of studies

Electronic search results were imported into EndNote X9 (Clarivate Analytics, Philadelphia, PA, USA). Subsequently, all duplicate records were deleted according to the methodology described by Bramer *et al.* [13]. An additional search in the reference lists of the studies included was carried out to identify other publications not identified by the systematic search. Duplicate records were evaluated by title and abstract to verify compliance with the inclusion criteria by two independent reviewers (MTYA and MLAM) using the Rayyan web application (https://rayyan.qcri.org/) [14]. Records meeting the inclusion criteria were evaluated in full text by the same two reviewers to determine their inclusion. Any disagreement at the end of the selection procedure was resolved between the two reviewers and if a decision could not be made, a third reviewer participated in the discussion (DA).

## Data extraction process

An *ad-hoc* form prepared by the authors in the Microsoft Excel program (Microsoft Corporation, Redmond, Washington, USA) was used for the extraction of data from the studies included in the systematic review. The information was independently extracted by two reviewers (MTYA and MLAM). Subsequently, a third reviewer conducted an assessment of the information extracted from 10 randomly selected studies. In case of inconsistencies, the two reviewers (MTYA and MLAM) were informed to re-evaluate the information and correct inconsistencies.

The articles included sought the reporting of information related to the presence/absence and/or severity of oral pathologies. The oral pathologies considered were tooth decay, malocclusion, TDI, periodontal diseases, temporomandibular dysfunction, salivary gland pathologies, cleft lip and palate, and edentulism. The outcome of interest was the OHRQoL report evaluated by specific instruments such as: CPQ11-14, ECOHIS, ECOHIS-B, OHIP-14, etc. In addition, the general characteristics of each study were extracted, such as: country in which the study was carried out, year of publication, first author of the study, journal, title, study design,

age group of the evaluated sample, OHRQoL instrument used, diagnostic indices of oral diseases, and language of publication, among others.

## Synthesis of results

A formal narrative synthesis of the collected data was performed. The synthesis focused mainly on the qualitative analysis of OHRQoL pathologies and results, separated by country of origin, age group evaluated, OHRQoL instrument used, among others. The limitations of the studies included were discussed.

For the assessment of the impact of oral diseases on OHRQoL, the results of differences in average quality of life were reported based on the presence/absence and/or severity of oral diseases, as well as the association of the strength of the results in a crude or adjusted manner, such as: relative risks (RR), odds ratios (OR) and prevalence ratios (PR). In cases in which any researcher disagreed with respect to the extracted data, the full-text article was revised again to verify the extracted data or to correct errors.

Due to the heterogeneity among the studies included, we considered a meta-analysis inappropriate, and therefore, we focused on the qualitative synthesis of the studies.

## Evaluation of study quality

Two authors (MTYA and MLAM) independently evaluated the quality of the studies using the U.S. National Institutes of Health tool [15]. This tool presents a different number of questions depending on the design of the study evaluated (14 questions for cohort or cross-sectional studies, and 12 for case-control studies). Each question has five possible answers: 'yes', 'no', 'cannot be determined', 'not applicable' and 'not determined'. After completing all the questions, each article was graded as good, fair, or poor quality. Disagreements were resolved by consensus with a third reviewer (DA).

## Ethical considerations

The completion of the study was approved by the Ethical Committee of the Institutional Policy of the *Universidad Científica del Sur* (Approval No. 010-2020-PRE15).

## Results

### Features of the studies included

Following the evaluation of 3310 articles by title and abstract, 46 articles were included for full-text evaluation, of which 40 articles were included in the systematic review (Fig 1). Table 1 provides an overview of the 40 studies included. Of these, 36 (90%) were carried out in Brazil [16–51], one was from Peru (2.5%) [52], another study was from Chile (2.5%) [53], one study was from Trinidad and Tobago (2.5%) [54] and one other study was performed in Colombia (2.5%) [55]. In relation to the study design, 32 were cross-sectional studies, 2 were cohort studies and there were 6 case and control studies. With regard to the classification of studies included according to age group, 24 studies were performed in children [16–19, 25, 27, 29–37, 39, 40, 43, 45, 47, 51, 52, 54, 55], 10 were in adolescents [20, 22–24, 26, 28, 44, 46, 48, 50], 4 were adult studies [21, 38, 42, 53] and 2 study included more than one age group [41, 49]. They all also evaluated the impact of oral disease over OHRQoL, classified according to exposure to one or more oral diseases, tooth decay, malocclusion, TDI, periodontal diseases, temporomandibular dysfunction, salivary gland pathologies, cleft lip and palate, and edentulism and found studies related to dental caries (n = 7) [20, 25, 32, 36, 37, 43, 54], malocclusion (n = 2) [24, 26], TDI (n = 3) [29, 39, 44], periodontal disease (n = 1) [38], cleft lip and palate

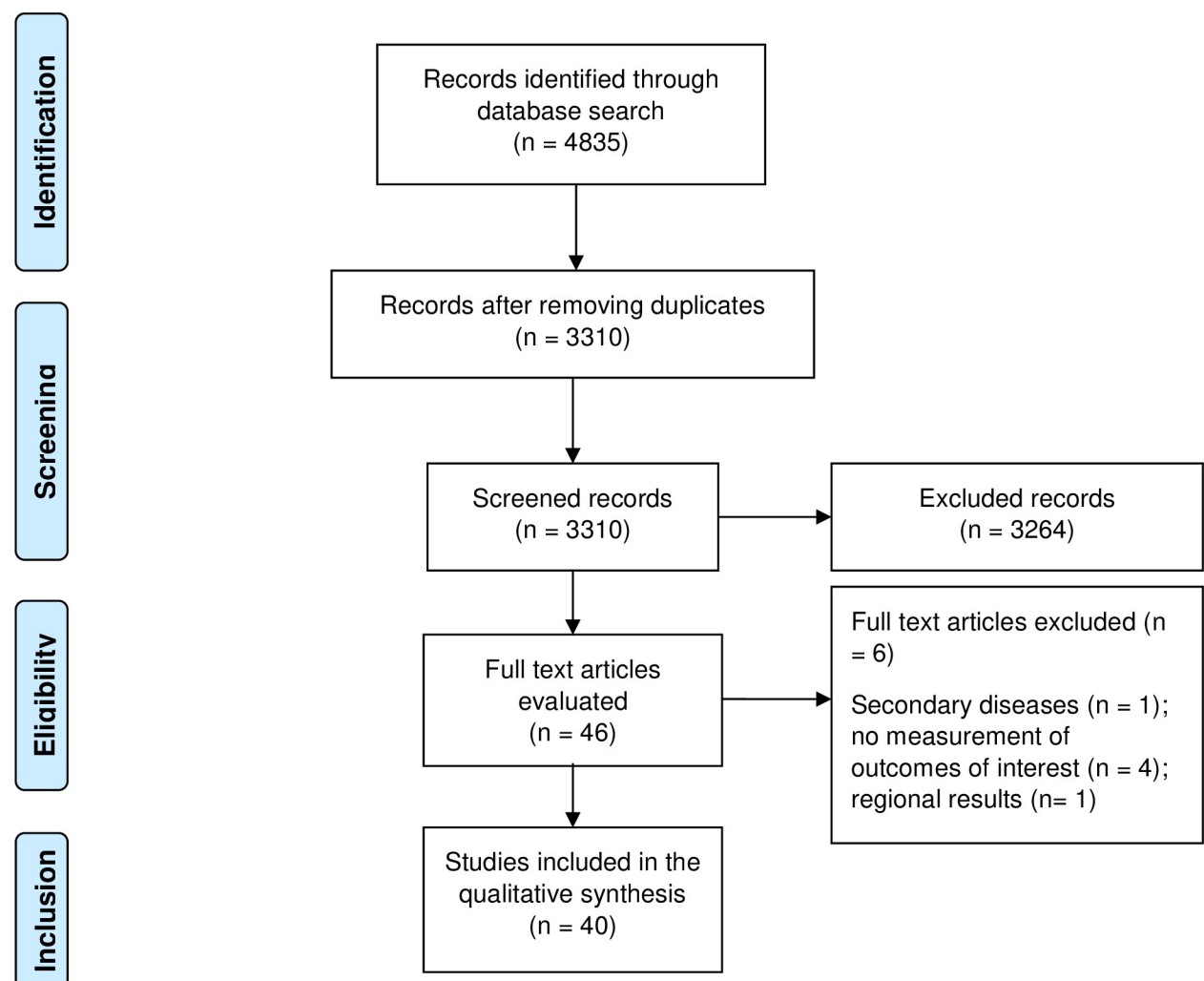

**Fig 1. Flow chart of study selection according to the PRISMA statement.**

(n = 1) [40], temporomandibular dysfunction (n = 1) [41], edentulism (n = 1) [46], salivary glands disease (n = 1) [53] and more than one oral disease evaluated (n = 23) [16–19, 21–23, 27, 28, 30, 31, 33–35, 42, 45, 47–52, 55].

The quality of life measurement instruments used in the different studies were as follows; CPQ8-10 [27, 31, 35, 36, 40, 47, 49], CPQ11-14 [20, 22–24, 26, 28, 48–50], ECOHIS [16, 29, 33, 37, 45, 52, 54], B-ECOHIS [18, 19, 25, 30, 32, 39, 43, 51],OHIP-14 [21, 42, 46, 53],OHIP [38, 41],C-ECOHIS [55], SOHO-5 [17], B-SOHO-5 [34] and Child-OIDP [44], the most frequent being CPQ11-14(n-9), B-ECOHIS(n-8). The association between quality of life and oral diseases was expressed using RR, OR and PR and mean difference hypothesis tests and beta coefficients.

## Tooth decay and OHRQoL

Thirty studies investigating the relationship between tooth decay and OHRQoL were identified (Table 1). Twenty-seven studies were conducted in Brazil, one in Peru, one in Trinidad and Tobago and one in Colombia. Twenty-eight studies were developed in populations under

**Table 1. Characteristics of the studies included.**

| N | Author/Year | Country (Publication language) | Study Design | Sample Size | Age group | Oral diseases (Instrument) | Instrument OHRQoL |
|---|---|---|---|---|---|---|---|
| 1 | Alves et al. 2013 | Brazil (English) | Cross-sectional | 1528 | 12 years | Dental caries (DMFT) | CPQ11-14 |
| 2 | Tomazoni et al. 2014 | Brazil (English) | Cross-sectional | 1134 | 12 years | Periodontal diseases (CPI); TDI (O'Brien classification); Malocclusion (DAI); Dental caries (DMFT) | CPQ11-14 |
| 3 | Da Rosa et al. 2016 | Brazil (English) | Cross-sectional | 1134 | 12 years | Malocclusion (DAI) | CPQ11-14 |
| 4 | López Ramos & García Rupaya. 2014 | Peru (Spanish) | Cross-sectional | 153 | 3 to 5 years | Dental caries (dmft); Malocclusion (Presence/Absence)*; TDI (Andreasen & Andreasen classification) | ECOHIS |
| 5 | Piva et al. 2018 | Brazil (English) | Cohort | Baseline: 163; two-year follow-up: 119 | 3 to 4 years | Dental caries (ICDAS criteria) | B- ECOHIS |
| 6 | Feldens et al. 2016 | Brazil (English) | Cross-sectional | 509 | 11 to 14 years | Dental caries (DMFT); malocclusion (DAI); TDI (Andreasen & Andreasen classification) | CPQ 11–14 Brazilian version |
| 7 | Martins et al. 2017 | Brazil (English) | Case-control | 546 (182 cases and 364 controls) | 8 to 10 years | Dental caries (DFMT/dmft); Malocclusion (DAI); TDI (Andreasen & Andreasen classification) | CPQ8-10 |
| 8 | Feldens et al.2016 | Brazil (English) | Cross-sectional | 1275 | 1 to 5 years | TDI (Andreasen & Andreasen classification) | ECOHIS |
| 9 | Oliveira de Lima et al. 2016 | Brazil (English) | Cross-sectional | 101 | 16 to 80 years | Temporomandibular dysfunction (TMI) | OHIP |
| 10 | Kramer et al. 2013 | Brazil (English) | Cross-sectional | 1036 | 2 to 5 years | Dental caries (dmft); Malocclusion (Presence/Absence)*; TDI (Andreasen & Andreasen classification). | ECOHIS |
| 11 | Abanto et al. 2014 | Brazil (English) | Cross-sectional | 335 | 5 to 6 years | Dental Caries (dmft); TDI (classified according to Glendor et al.) | SOHO |
| 12 | Martins-Júnior et al. 2013 | Brazil (English) | Cross-sectional | 638 | 2 to 5 years | Dental caries (dmft) | ECOHIS |
| 13 | Simões et al. 2017 | Brazil (English) | Cross-sectional | 1206 | 8 to 12 years | Dental caries (DMFT); Malocclusion (DAI); TDI (O´Brien criteria) | CPQ8-10 and CPQ11–14 |
| 14 | Dutra et al. 2018 | Brazil (English) | Cross-sectional | 270 | 8 to 10 years | Dental caries (DMFT/deft); Malocclusion (DAI) | CPQ8-10 |
| 15 | Abanto et al. 2011 | Brazil (English) | Cross-sectional | 260 | 2 to 5 years | Dental caries (dmft); Malocclusion (presence/absence)*; TDI (Andreasen & Andreasen classification) | ECOHIS |
| 16 | Firmino et al. 2016 | Brazil (English) | Case-control | 415 | 3 to 5 years | Dental caries (ICDAS-II); TDI (presence/absence) | B-ECOHIS |
| 17 | Meusel et al. 2015 | Brazil (English) | Cross-sectional | 100 | 30 to 58 years | Periodontal diseases (mild/moderate/severe) | OHIP-14Br |
| 18 | Rocha et al. 2016 | Brazil (English) | Cross-sectional | 224 | 12 years | Edentulism (presence/absence)* | OHIP-14 |
| 19 | Batista et al. 2014 | Brazil (English) | Cross-sectional | 248 | 20 to 64 years | Dental caries (DMFT); Edentulism (number of missing teeth) | OHIP-14 |
| 20 | Abanto et al. 2015 | Brazil (English) | Cross-sectional | 1215 | 1 to 4 years | Dental caries (dmft); Malocclusion (presence/absence)^l; TDI (Classified according to Glendor et al.) | B-ECOHIS |
| 21 | Aldrigui et al. 2011 | Brazil (English) | Cross-sectional | 260 | 2 to 5 years. | Dental caries (dmft); Malocclusion (presence/absence)*; TDI (Classified according to Glendor et al.) | B-ECOHIS |
| 22 | Milani et al. 2019 | Brazil (English) | Cross-sectional | 146 | 2 to 6 years | TDI (Andreasen & Andreasen classification) | B-ECOHIS |
| 23 | Vieira-Andrade et al. 2015 | Brazil (English) | Case-control | 335 (67 cases and 268 controls) | 3 to 5 years | Dental caries (ICDAS); TDI (Andreasen & Andreasen classification); malocclusion (presence/absence)* | B-ECOHIS |
| 24 | Ramos-Jorge et al. 2014 | Brazil (English) | Cross-sectional | 668 | 12 to 15 years | TDI (O´Brien classification) | Child-OIDP |
| 25 | Ramos-Jorge et al. 2014 | Brazil (English) | Cross-sectional | 499 | 3 to 5 years | Dental caries (ICDAS, ALA); Malocclusion (presence/absence)*; TDI (Andreasen & Andreasen classification). | ECOHIS |
| 26 | Corrêa-Faria et al. 2018 | Brazil (English) | Cross-sectional | 563 | 2 to 5 years | Dental caries (dmft and Pufa index) | B- ECOHIS |
| 27 | Guedes et al. 2016 | Brazil (English) | Cohort | Baseline: 487; two year follow-up: 352 | 1 to 5 years | Dental caries (ICDAS) | B- ECOHIS |

*(Continued)*

**Table 1.** (Continued)

| N | Author/Year | Country (Publication language) | Study Design | Sample Size | Age group | Oral diseases (Instrument) | Instrument OHRQoL |
|---|---|---|---|---|---|---|---|
| 28 | Sardenberg et al. 2013 | Brazil (English) | Cross-sectional | 1204 | 8 to 10 years | Dental caries (dmft); Malocclusion (DAI) | CPQ8-10 |
| 29 | Scapini et al. 2013 | Brazil (English) | Cross-sectional | 509 | 11 to 14 years | Dental caries (DMFT); Malocclusion (DAI); TDI (presence/absence)* | CPQ11–14/ ISF:16 |
| 30 | Bittencourt et al. 2017 | Brazil (English) | Cross-sectional | 1614 | 11 to 14 years | Dental caries (DMFT); Malocclusion (DAI); TDI (Andreasen classification) | CPQ11-14/ ISF:16 |
| 31 | Naidu et al. 2016 | Trinidad and Tobago (English) | Cross-sectional | 245 | 3 to 5 years | Dental Caries (dmft) | ECOHIS |
| 32 | Bendo et al. 2014 | Brazil (English) | Case-control | 1215 (405 cases and 810 controls). | 11–14 years | Dental caries (World Health Organization recommendations); Malocclusion (DAI); TDI (Andreasen classification) | CPQ11-14-ISF:16 |
| 33 | Freire-Maia et al. 2015 | Brazil (English) | Cross-sectional | 1201 | 8 to 10 years | Dental caries (World Health Organization criteria); TDI (Andreasen classification); Malocclusion (DAI) | CPQ8-10 Brazilian version. |
| 34 | Montes et al. 2019 | Brazil (English) | Case-control | 108 | 8 to10 years | Unilateral cleft lip and palate (Presence/absence)*. | CPQ8-10 Brazilian version. |
| 35 | Díaz et al. 2018 | Colombia (English) | Cross-sectional | 643 | 1 to 5 years | Dental caries (dmft); Malocclusion (Presence/absence)*; TDI (Andreasen classification) | C-ECOHIS |
| 36 | Maia et al. 2019 | Brazil (English) | Cross-sectional | 238 | 5 to 6 years | Dental caries (dmft); TDI (Presence/absence)* | B-SOHO 5 |
| 37 | Niklander et al. 2017 | Chile (English) | Case-control | 566 | 18 to 83 years | Salivary gland pathology: xerostomia (VAS) | OHIP-14 sp |
| 38 | Bretz et al. 2019 | Brazil (English) | Cross-sectional | 117 | 11 to 12 years | Malocclusion (DAI) | CPQ11-14 |
| 39 | Passos-Soares et al. 2018 | Brazil (English) | Cross-sectional | 306 | 18 to 80 years | Dental caries (DMFT); Periodontal disease (periodontal status) | OHIP-14 |
| 40 | Martins-Júnior et al. 2012 | Brazil (English) | Cross-sectional | 112 | 8 to 10 years | Dental caries (DMFT or dmft) | CPQ8-10 |

*The diagnostic instrument used was not specified.

TDI: Traumatic Dental Injuries; DMFT: decayed/missing/ filled for permanent teeth; dmft: decayed, missing and filled teeth index for primary teeth.

B-ECOHIS: Brazilian version of the Early Childhood Oral Health Impact Scale.

C-ECOHIS: Colombian version of the Early Childhood Oral Health Impact Scale.

ECOHIS: Early Childhood Oral Health Impact Scale.

Child-OIDP: Child Oral Impact on Daily Performances.

SOHO 5: Brazilian version of the Scale of Oral Health Outcomes for 5-years.

B-SOHO 5: Brazilian version of the Scale of Oral Health Outcomes for 5-years.

SOHO: Scale of Oral Health Outcomes.

CPQ 8–10: The Child Perceptions Questionnaire for children aged 8 to 10 years.

CPQ11-14: The Child Perceptions Questionnaire for children aged 11 to 14 years.

CPQ11-14/ISF:16: The Child Perceptions Questionnaire for children aged 11 to 14 years–Impact Short Form.

OHIP.14sp: Spanish version of the oral health impact profile-14 questionnaire.

OHIP-14: Oral Health Impact Profile for 14 years.

OHIP-14Br: Brazilian version of the Oral Health Impact Profile for 14 years.

DAI: Dental Aesthetic Index VAS: Visual Analogue Scale.

ICDAS: International Caries Detection and Assessment System.

PUFA: Pulpal, Ulceration, Fistula, Abscess, index.

ALA: Activity Lesion Assessment.

CPI: Community periodontal index criteria.

the age of 18, while two studies included a population of 18–80 years [21, 42]. To assess the presence of tooth decay in the population, 9 studies used the DMFT index, 11 the dmft index, 5 the ICDAS scale, 1 activity injury assessment (ALA), 2 ranked according to the WHO, 1 Pufa index, and 1 study used DMFT/dmft. In addition, 3 studies [25, 27, 45] used more than one instrument to evaluate tooth decay. Of the 30 studies evaluated (Table 2), only three showed no significant association between the presence of tooth decay and OHRQoL [20, 22, 51]. One of the studies included reported that tooth decay affected OHRQoL only when associated with periodontitis [42]. In addition, Correa-Faria *et al*. [25] reported that only cavities with clinical consequences are associated with poor OHRQoL. On the other hand, Simões *et al*. [49] reported a significant association between tooth decay and OHRQoL in children between the age of 8 and 10 compared to pre-adolescents between the age of 11 and 14.

## Malocclusion and OHRQoL

Twenty studies investigating the association between malocclusion and OHRQoL were identified (Table 1), 18 in Brazil, one in Peru and one in Colombia. Of these, 12 studies included children between 2 and 10 years old [16, 18, 19, 27, 31, 33, 35, 45, 47, 51, 52, 55], 7 included the pre-adolescent population between 11 and 14 years old [22–24, 26, 28, 48, 50], and one was in children between 8 and 12 years old [49]. Of the total studies, 12 assessed malocclusions according to the criteria of the dental aesthetic index (DAI) while 9 evaluated the absence or presence of the disease. As for the results obtained by these 20 studies (Table 2), 9 studies reported a significant negative impact of malocclusion on OHRQoL [22, 26, 27, 31, 33, 47–50], 11 did not describe a significant association between this pathology and OHRQoL. One of the studies [23] found no significant association between severe malocclusion and OHRQoL, except in cases of definite and handicapping malocclusion. In addition, two studies reported that the most severe states of malocclusion were associated with worse OHRQoL, even though the less severe states of the condition were not associated [24, 49].

## TDI and OHRQoL

Of the 22 studies on TDI evaluated (Table 1), 20 were conducted in Brazil, 1 in Peru and one in Colombia. Fifteen of the 22 articles identified investigated children between the ages of 2 and 10, while 6 were on adolescents between the ages of 11 and 14 and one studied children between the ages of 8 and 12. To evaluate TDI, the following instruments were used: 13 articles used Andreasen's classification [16, 22, 23, 28, 29, 31, 33, 35, 39, 45, 51, 52, 55], 3 the O'Brien classification [44, 49, 50], 3 [17–19] the Glendor *et al*. [56] and 3 did not indicate which instrument was used for the diagnosis of TDI [30, 34, 48]. Of the total studies, 13 found a significant negative association between TDI and OHRQoL (Table 2). Nine studies [16, 23, 28, 35, 48–52] reported that quality of life was not influenced by the presence of TDI.

## Periodontal disease and OHRQoL

Three studies in Brazil evaluated the association between periodontal disease and OHRQoL. (Table 1). One study [50] included 12-year-olds, while two studies [38, 42] included adult populations. The Tomazoni *et al*. study evaluated the impact of gingival bleeding in children on OHRQoL, as measured with the CPQ 11–14 tool. It was concluded that a degree of bleeding ≥15% increased the risk of having a worse OHRQoL, and this association was maintained even after adjustment according to clinical, social demographic and economic variables. In their study, Meusel *et al*. evaluated the impact of the severity of periodontal disease on OHR-QoL in adults measured with the OHIP-14Br tool, and concluded that individuals with severe periodontal disease had a worse OHRQoL than those with mild/moderate periodontal disease

**Table 2. Summary of results of the studies of different oral diseases and OHRQoL.**

| N | Author/Year | Oral diseases | Reference/Exposure* | Primary Outcome | Main findings | Crude or Adjusted |
|---|---|---|---|---|---|---|
| 1 | Alves et. al. 2013 | Dental caries | Caries intraoral distribution: Caries free/Only posterior teeth/ Anterior teeth | CPQ11–14 | aRR (95% CI) *p<0.05: Only posterior teeth: 1.01 (0.96–1.06)/ Anterior teeth 1.11 (0.99–1.24). | Gender and socio-economic status |
| 2 | Tomazoni et al. 2014 | Periodontal diseases/ Dental caries/TDI Malocclusion | Gingival bleeding: <15% sites/> = 15% sites Dental caries: Without/With Dental trauma: Without/With Malocclusion: Without/With | CPQ11-14 | aRR (95% CI) *p<0.05. Gingival bleeding: 1.15 (1.10 to 1.21)* Dental caries: 1.08 (1.03 to 1.12)* Dental Trauma: 1.00 (0.94 to 1.04) Malocclusion: 1.16 (1.11 to 1.21)* | Clinical oral conditions and broad individual and contextual-level covariates. |
| 3 | Da Rosa et al. 2016 | Malocclusion | Severity of Malocclusion: None-minor / Definite/ Severe/ Handicapping | CPQ11-14 | aRR (95% CI) *p<0.001: Malocclusion: Definite: 1.07 (1.01–1.12)*/ Severe: 1.20 (1.11–1.28)*/ Handicapping: 1.26 (1.17–1.35)*. | Mother's education, father's education, dental caries, skin color, gender, dental trauma; and mean income of the neighborhood |
| 4 | López Ramos et. al. 2014 | Dental caries/ Malocclusion/TDI | Early Childhood Caries: caries free/low severity/high severity. TDI: absence/presence. Anterior malocclusion: absence/presence. | ECOHIS | Mean (SD) *p<0.05: Caries free: 8.75 (7.33)*/ Low severity: 13.78 (8.28)*/ High severity: 24.88 (9.43)*. TDI: Absence: 15.22 (10.42)/ Presence: 18.07 (8.22). Anterior malocclusion: Absence: 15.21 (9.97)/ Presence: 19.6 (13.51). | Crude |
| 5 | Piva, F et al. 2018 | Dental caries | Early childhood caries: dmf-t = 0/ dmf-t $\geq$1 | B-ECOHIS | cRR (95% CI) *p<0.05: At baseline: dmf-t $\geq$1: 2.75 (1.33–5.68)*. 2 years follow up: dmf-t $\geq$1: 3.12 (1.22–7.96)* | Crude |
| 6 | Feldens et. al. 2016 | Dental caries/ Malocclusion/TDI | Dental caries: Caries free/ low severity/ high severity. Malocclusion: absence/ presence. TDI: absence/ presence. | CPQ 11–14 Brazilian version | aRR (95% CI) *p<0.05: Dental caries: low severity: 1.11 (0.98–1.27)/ high severity: 1.30 (1.12–1.51)*. Malocclusion: presence: 1.13 (1.00–1.28). cRR (95% CI) *p< 0.05: TDI: presence: 1.06 (0.87–1.29). | Gender, ethnic group, monthly household income, family structure and mother's schooling. Unadjusted for TDI. |
| 7 | Martins et. al. 2017 | Dental caries/ Malocclusion/TDI | Dental caries: DMFT/dmft = 0/ DMFT/dmft = 1 or 2/ DMFT/ dmft $\geq$ 3. Malocclusion: absent or mild/ present. TDI: absent/ present | CPQ8-10 | aOR (95% CI) *p<0.05: DMFT/ dmft = 1 or 2: 1.61 (1.05–2.49)*/ DMFT/dmft $\geq$ 3: 2.06 (1.28–3.31)*. Malocclusion: present: 1.21 (0.82–1.78). TDI: Present: 1.11 (0.66–1.87). | Adjusted (Not specified) |
| 8 | Feldens et al. 2016 | TDI | TDI: Absence/ Enamel fracture/ Other TDI | ECOHIS | aPR (95% CI) *p<0.05: Enamel fracture: 1.10 (0.62–1.93)/ Other TDI: 1.87 (1.39–2.52)* | Age, malocclusion, use of dental services, and dental caries |
| 9 | Oliveira de Lima et al. 2016 | Temporomandibular Dysfunction | TMD: Muscle Disorder: No dysfunction/dysfunction. Disc Displacement: No dysfunction/ dysfunction. Arthralgia, arthritis, and arthrosis: No dysfunction/ dysfunction. | OHIP-14 | Mean (SD) *p<0.05: TMD: Muscle Disorder: No dysfunction: 7.32(3.99)/ dysfunction: 11.76(6.27)*. Disc Displacement: No dysfunction: 10.04(6.15)/ dysfunction: 10.34 (5.87). Arthralgia, arthritis and arthrosis: No dysfunction: 10.05 (5.44)/ dysfunction: 10.33(6.30). | Crude |
| 10 | Kramer et. al. 2013 | Malocclusion/ Dental Caries/ TDI | Dental caries: Caries free/ low severity/ high severity. TDI: Absent/ present. Malocclusion: absent/ present. | ECOHIS | aRR (95% CI) *p<0.05: Caries dental: Low severity: 1.62 (1.17–2.23)*/ High severity: 2.74 (2.02–3.72)*. TDI: Presence: 1.70 (1.27–2.27)*. Malocclusion: Presence: 1.42 (1.04–1.94)* | Malocclusion: Age, Dental Caries, TDI: Age, malocclusion. |

(*Continued*)

**Table 2.** (Continued)

| N | Author/Year | Oral diseases | Reference/Exposure* | Primary Outcome | Main findings | Crude or Adjusted |
|---|---|---|---|---|---|---|
| 11 | Abanto et. al. 2014 | Dental caries/TDI | Dental caries: Caries free/ low experience of caries/ high experience of caries. TDI: Absence/ Uncomplicated/ complicated. | SOHO | aRR (95% CI) *p<0.05: Child report: Dental caries: Low experience: 3.85 (2.83–5.23)*/ High experience: 6.37 (4.71–8.62)*. cTDI: Uncomplicated: 0.69 (0.51–0.94)*/Complicated: 1.11 (0.84–1.47). aRR (95% CI) *p<0.05: Parental reports: Low experience: 4.82 (3.39–6.85)*/ High experience: 10.81 (7.65–15.27)*. cTDI Uncomplicated: 0.83 (0.57–1.22)/Complicated: 1.10 (0.81–1.50). | Gender, age, family income/ Crude for TDI. |
| 12 | Martins-Júnior et al. 2013 | Dental Caries | Dental caries: Caries free/ low severity/ high severity | ECOHIS | aRR (95% CI) *p<0.05: Low severity: 2.45 (1.64–3.65)*/ High severity: 5.32 (3.67–7.71)* | Age of the child, age of the mother, gender. |
| 13 | Simões et al. 2017 | Malocclusion/TDI/ Dental caries | Malocclusion: Normal or Mild/ Definite/ Severe / Very severe. TDI: Absent/ present. Dental caries: Absent/ Present | CPQ8-10 and CPQ11-14 | aRR (95% CI) *p<0.05 CPQ8-10: Malocclusion: Definite: 1.10 (0.95–1.27)/ Severe: 1.04 (0.87–1.24)/ Very severe: 1.24 (1.02–1.51)*. TDI: Presence: 1.23 (0.89–1.70). Dental caries: Present: 1.16 (1.03–1.32)* aRR (95% CI) *p<0.05 CPQ11-14: Malocclusion: Definite: 1.09 (0.90–1.32)/Severe: 1.17 (0.89–1.54)/ Very severe: 1.28 (1.01–1.62)*. Dental trauma: Severe: 1.02 (0.71–1.47). Dental caries: Present: 1.14 (0.98–1.35). | Adjusted for demographic variables (Gender, age, and race), socioeconomic characteristics (maternal education and family income), clinical variables (dental caries and dental trauma). |
| 14 | Dutra et al. 2018 | Malocclusion/ Dental caries | Malocclusion: Extremely severe malocclusion Normal or Mild/ Definite/ Severe Dental caries: Untreated decayed teeth: absence/ presence. | CPQ8-10 | cPR (95% CI) *p<0.05 Malocclusion: Normal or Mild: 0.442 (0.258–0.758)*/ Definite: 0.591 (0.339–1.031)/ Severe: 0.407 (0.224–0.740)* Dental caries: cPR (95% CI) *p<0.05: Untreated decayed teeth: absence: 0.758 (0.613–0.937)* | Crude |
| 15 | Abanto et al. 2011 | Dental caries/TDI/ Malocclusion | Dental caries: Caries free/ low severity/ high severity. TDI: Absent/ present. Malocclusion: absent/ present | ECOHIS | aRR (95% CI) *p<0.05: Dental caries: Low severity: 2.03 (1.40–2.95)*/ High severity: 3.89 (2.68–5.64)*. aRR (95% CI) *p<0.05: TDI: Presence: 1.27 (0.98–1.65). Malocclusion: Presence: 0.95 (0.73–1.24). | Family income |
| 16 | Firmino et al. 2016 | Dental caries/TDI | Dental caries: No cavitated lesion and/or with white spots/Low severity/ High severity. Absence/ Present TDI: Absence/ presence. | B-ECOHIS | aOR (95% CI) *p<0.05: Dental caries: Low severity: 2.98 (1.69–5.26)*/ High severity: 12.58 (5.31–29.79)*. TDI: Presence: 2.11 (1.23–3.62)*. cOR (95% CI) *p<0.05: Dental caries: Present: 3.66 (2.14–6.24)*. | Type of preschool, parent's/ caregiver's perception of child's general health, parent's/ caregiver's perception of child's oral health, and history of dental visit/ Crude for present or absent dental caries. |
| 17 | Meusel et al. 2015 | Periodontal diseases | Periodontal disease severity: Mild or moderate chronic periodontitis/Severe chronic periodontitis | OHIP- 14Br | Mean (SD) *p<0.05: Mild/ moderate 18.2 (12.9)/ Severe: 24.1 (14.8)* | Crude |

(*Continued*)

**Table 2.** (Continued)

| N | Author/Year | Oral diseases | Reference/Exposure* | Primary Outcome | Main findings | Crude or Adjusted |
|---|---|---|---|---|---|---|
| 18 | Rocha et al. 2016 | Edentulism | Loss of anterior teeth: No/Yes | OHIP-14 | Mean (SD) *p<0.05: Loss of anterior teeth: No: 6.28 (5.85)/ Yes: 8.43 (6.70)* | Crude |
| 19 | Batista et al. 2014 | Edentulism/Dental caries | Tooth loss: No teeth lost/ Edentulous/Lost 13–31 teeth/ Loss of up to 12 teeth, including 1 + anterior/ Loss of up to 12 posterior teeth, excluding 1st molars/ Loss of 1–4 1st molar. Dental caries: Yes/No. | OHIP-14 | aOR (95% CI) *p<0.05: Tooth loss: Edentulous: 3.92 (0.94–16.89)/ Lost 13–31 teeth: 1.08 (0.38–3.05)/ Loss of up to 12 teeth, including 1+ anterior: 0.96 (0.37–2.51)/ Loss of up to 12 posterior teeth, excluding 1st molars: 1.37 (0.38–4.97)/ Loss of 1 to 4 1st molar: 1.28 (0.30–5.52). Caries: Yes: 3.96 (1.85–8.51)* | Socioeconomic factors, dental care use, smoking, untreated caries, and oral health literacy. |
| 20 | Abanto et al. 2015 | TDI/Malocclusion/ Dental caries | TDI severity: Absence/ Uncomplicated injuries/ Complicated injuries. Malocclusion: Absence/ presence. Dental caries: Absence/ presence. | B-ECOHIS | aPR (95% CI) *p<0.05: TDI severity: Uncomplicated injuries: 0.75 (0.55–1.03)/Complicated injuries: 2.10 (1.01–4.39)*. Malocclusion: Presence: 0.87 (0.67–1.13). Dental caries: Presence: 3.09 (2.28–4.20)* | Age, dental caries |
| 21 | Aldrigui et al. 2011 | TDI/Malocclusion/ Dental caries | TDI severity: Absence/ Uncomplicated injuries/ Complicated injuries. Malocclusion: absence/ presence. ECC: caries free (dmf-t = 0)/ low severity (dmf-t = 1–5)/ high severity (dmf-t ≥6) | B-ECOHIS | aRR (95% CI) *p<0.05: TDI severity: Uncomplicated: 0.89 (0.66–1.20)/ Complicated: 1.90 (1.38–2.62)*. Malocclusion/ presence: 0.97 (0.75–1.26). ECC: Low severity: 2.02 (1.39–2.92)*/ high severity: 4.23 (2.96–6.05)* | Severity of ECC |
| 22 | Milani et al. 2019 | TDI | TDI: uncomplicated/ complicated | B-ECOHIS | Mean (SD) *p<0.05: TDI: uncomplicated: 6.14 (7.27)/ complicated: 9.34 (8.41)* | Crude |
| 23 | Vieira-Andrade et al. 2015 | TDI/Malocclusion/ Dental caries | TDI: absence/ Presence. Malocclusion: absence/ presence. Dental caries: dmft = 0/ dmft≥1 | B-ECOHIS | aOR (95% CI) *p<0.05: TDI:/ Presence: 1.15 (0.66–2.03). Malocclusion: Presence: 1.01 (0.57–1.78). Dental caries: dmft≥1: 1.14 (0.65–1.99). | Dental caries and malocclusion |
| 24 | Ramos-Jorge et al. 2014 | TDI | TDI: Child-OIDP = 0/ Child-OIDP ≥1 | Child-OIDP | aPR (95% CI) *p<0.05: Child-OIDP ≥1: 1.73 (1.2–2.4)* | Age, gender, mother's schooling, malocclusion, and dental caries |
| 25 | Ramos-Jorge et al. 2014 | Dental caries/ Malocclusion/TDI | Dental caries: Distinct cavity with visible dentin: Active = 0/ ≥1/ Extensive cavity: Without pulp exposure: Active: = 0/ ≥1; Inactive: = 0/ ≥1. With pulp exposure and absence of fistula: = 0/ ≥1. Root remnant.: = 0/≥1 TDI: Absence/ presence. Malocclusion: Absence/ presence. | ECOHIS | aRR (95% CI) *p<0.05: Dental caries: Distinct cavity with visible dentin: Active ≥1: 1.74 (1.29–2.34)*. Extensive cavity: Without pulp exposure: Active ≥1: 4.28 (3.05–6.01)*. Inactive ≥1: 3.68 (1.74–7.81)*. With pulp exposure and absence of fistula: ≥1: 1.57 (1.16–2.12)*. Root remnant: ≥1: 1.52 (1.17–1.98)*. TDI: Presence: 1.70 (1.26–2.28)*. cRR (95% CI) *p<0.05: Malocclusion: cRR (95% CI) Presence: 1.00 (0.71–1.41) | Educational level of mother, easy access to medical care, child's age/ Crude for malocclusion. |

(*Continued*)

**Table 2.** (Continued)

| N | Author/Year | Oral diseases | Reference/Exposure* | Primary Outcome | Main findings | Crude or Adjusted |
|---|---|---|---|---|---|---|
| 26 | Corrêa-Faria et al. 2018 | Dental Caries | Untreated dental caries: Without dental caries/ without clinical consequences/ with clinical consequences. | B-ECOHIS | aPR (95% CI) *p<0.05: Untreated dental caries: Without clinical consequences: 1.16 (0.99–1.35)/ With clinical consequences: 1.31 (1.01–1.70)* | Age, gender, mother's schooling |
| 27 | Guedes et al. 2016 | Dental caries | Dental caries: No caries lesions/ Only initial caries lesions/ At least one moderate caries lesion/ Extensive lesions | B-ECOHIS | aRR (95% CI) *p<0.05: Only initial caries lesions: 1.34 (0.77–2.36)/ At least one moderate caries lesion: 2.54 (1.17–5.50)*/ Extensive lesions: 4.28 (2.54–7.20)* | Household income |
| 28 | Sardenberg et al. 2013 | Malocclusion/Untreated dental caries | Malocclusion: absence/ presence. Untreated dental caries: absence/ presence | CPQ8-10 | aPR (95% CI) *p<0.05: Malocclusion: presence: 1.30 (1.15–1.46)* cPR (95% CI) *p<0.05: Untreated dental caries:: presence: 1.60 (1.35–1.88)*. | Dental caries, gender, social vulnerability, type of school, parents/guardians schooling, and number of people in household/ Crude for dental caries |
| 29 | Scapini et al. 2013 | Malocclusion/TDI/ Dental caries | Malocclusion: Normal or minor malocclusion/ Definite malocclusion/ Severe malocclusion/ Handicapping malocclusion. Dental trauma: absence/ presence. Dental caries: DMFT 0/ DMFT 1-3/ DMFT>3. | CPQ11–14/ ISF:16 | Mean (SD) *p<0.05: Malocclusion: Normal or minor: 11.75 (8.42)*/ Definite: 12.30 (8.57)*/ Severe: 13.39 (8.55)*/ Handicapping: 14.55 (8.97)*. Dental trauma: Absence: 12.75 (8.46)/ Presence: 13.54 (10.06). Dental caries: DMFT 0: 11.64 (8.54)/ DMFT 1–3: 13.35 (8.05)/ DMFT>3: 15.98 (9.18)*. | Crude |
| 30 | Bittencourt et al. 2017 | Malocclusion/TDI/ Dental caries | Malocclusion: Absent or mild malocclusion/ Definite/ Severe/ Handicapping. TDI: absent/ present. Dental caries: DMFT = 0/ DMFT≥1. | CPQ11-14-ISF:16 | aPR (95% CI) *p<0.05: Malocclusion: Definite: 1.11 (1.01–1.21)*/ Severe: 1.10 (0.98–1.25)/ Handicapping: 1.26 (1.13–1.42)*. cPR (95% CI) *p<0.05: TDI: present: 1.01 (0.93–1.10). cPR (95% CI) *p<0.01: Dental caries: DMFT≥1: 1.10 (1.03–1.17)* | Age, gender, social vulnerability index, DMFT, traumatic dental injuries and type of school. Crude for TDI and dental caries |
| 31 | Naidu et al. 2016 | Dental caries | ECC: dmft: dmft = 0/ dmft = 1-3/ dmft>4. | ECOHIS | aOR (95% CI) *p<0.05: Dental caries: dmft = 1–3: 2.55 (1.19–5.50)*/ dmft>4: 8.70 (3.54–23.13)* | Parental age/sex and child's age and sex |
| 32 | Bendo et al. 2014 | TDI/Dental caries/ Malocclusion | TDI: Without injuries/ Restored fracture/ Enamel fracture only/ Fracture involving dentin and or pulp. Dental caries: Without untreated lesion/ With untreated lesion. Malocclusion: absent or mild/ present | CPQ11-14-ISF:16 | aOR (95% CI) *p<0.05: TDI: Restored fracture: 1.75 (0.95–3.21)/ Enamel fracture only: 0.63 (0.39–1.03)/ Fracture involving dentin and or pulp: 2.40 (1.26–4.58)*. aOR (95% CI) *p<0.05: Dental caries: With untreated lesion: 1.30 (0.99–1.70). aOR (95% CI) *p<0.05 Malocclusion: Present: 1.68 (1.30–2.18)* | Age, malocclusion, and dental caries |
| 33 | Freire-Maia et al. 2015 | TDI/ Dental caries/ Malocclusion | TDI: Without or mild trauma/ Severe trauma. Dental caries: absence/ presence. Malocclusion: anterior maxillary overjet ≤3/ >4 | CPQ8-10 Brazilian version. | aOR (95% CI) *p<0.05: TDI: Severe trauma: 2.54 (1.21–5.31)*. Dental caries: Presence: 2.05 (1.50–2.80)* Malocclusion: Anterior maxillary overjet:/ >4: 1.64 (1.03–2.62)* | Gender, age, Parents/caregivers' level of education, TDI, dental caries, anterior maxillary overjet, type of school |

(*Continued*)

**Table 2.** (Continued)

| N | Author/Year | Oral diseases | Reference/Exposure* | Primary Outcome | Main findings | Crude or Adjusted |
|---|---|---|---|---|---|---|
| 34 | Montes et al. 2019 | Cleft lip and Palate (UCLP) | UCLP group/Control group | CPQ8-10 Brazilian version. | Mean (SD) *p<0.05: UCLP group: 17.2 (13.0)/ Control group: 13.4 (12.1) | Crude |
| 35 | Díaz et al. 2018 | Dental Caries/TDI/ Maloclusión | Caries: caries free/ low severity/ high severity. TDI: absence/ presence. Malocclusion: absence/ presence. | C-ECOHIS | aRR (95% CI) *p<0.05: Dental Caries severity: low severity: 1.53 (1.11–2.13)*/ high severity: 3.38 (2.35–4.86)*. TDI: Presence: 1.56 (1.16–2.11)*. Malocclusion: Presence: 1.09 (0.81–1.45) | Age, mother's education, household crowding, family structure/ Crude for malocclusion |
| 36 | Maia et al. 2019 | Dental caries/TDI | Dental caries: Caries free/ low caries experience/ high caries experience. TDI: Absence/ presence | B-SOHO 5 | Mean (SD) *p<0.05: Dental caries: Caries free: 0.94 (1.44)/ low caries experience: 1.75 (2.05)/ high caries experience: 4.61 (3.51)*. TDI: Absence: 2.14 (2.74)/ presence: 3.33 (3.24)* | Crude |
| 37 | Niklander et al. 2017 | Salivary gland pathology | With xerostomia/Without xerostomia | OHIP-14 | Mean (SD) *p<0.05: With xerostomia: 20.05 (14.32)/ Without xerostomia: 12.71 (12.99)* | Crude |
| 38 | Bretz et al. 2019 | Malocclusion | Malocclusion: Absent or slight/ Defined/ Severe | CPQ11-14 | aOR (95% CI) *p<0.05: Malocclusion: Defined: 1.54 (0.61–3.89)/ Severe: 2.63 (1.07–6.45)*. | Gender |
| 39 | Passos-Soares et al. 2018 | Dental caries/ periodontitis | Dental caries: Non exposed/ With caries or periodontitis/ With combined caries and periodontitis | OHIP-14 | aPR (95% CI) *p<0.05: Exposed to caries or periodontitis: 1.34 (0.87–2.05). Exposed combined to caries AND periodontitis: 1.63 (1.03–2.59)* | Age, sex, schooling level and smoking habit. |
| 40 | Martins-Júnior et al. 2012 | Dental caries | Dental caries: Without dental caries/ with untreated dental caries | CPQ8-10 | Mean (SD) *p<0.05: Dental caries: Without dental caries: 12 (7.1)/ with untreated dental caries: 23.1 (13.9)*. | Crude |

TDI: Traumatic Dental Injuries.

DMFT: decayed/missing/ filled for permanent teeth; dmft: decayed, missing and filled teeth index for primary teeth; SD: standard deviation; cOR: crude odds ratio; aOR: adjusted odds ratio; cPR: crude prevalence ratio; aPR: adjusted prevalence ratio, RR: relative risk; aRR: adjusted relative risk; cRR: crude relative risk; CI: confidence interval.

B-ECOHIS: Brazilian version of the Early Childhood Oral Health Impact Scale.

C-ECOHIS: Colombian version of the Early Childhood Oral Health Impact Scale.

ECOHIS: Early Childhood Oral Health Impact Scale.

Child-OIDP: Child Oral Impact on Daily Performances.

B-SOHO 5: Scale of Oral Health Outcomes for 5-years.

SOHO: Scale of Oral Health Outcomes.

CPQ 8–10: The Child Perceptions Questionnaire for children aged 8 to 10 years.

CPQ11-14: The Child Perceptions Questionnaire for children aged 11 to 14 years.

CPQ11-14/ISF:16: The Child Perceptions Questionnaire for children aged 11 to 14 years–Impact Short Form.

OHIP.14sp: Spanish version of the Oral Health Impact Profile-14 questionnaire.

OHIP-14: Oral Health Impact Profile for 14 years.

OHIP-14Br: Brazilian version of the Oral Health Impact Profile for 14 years.

DAI: Dental Aesthetic Index VAS: Visual Analogue Scale.

ICDAS: International Caries Detection and Assessment System.

PUFA: Pulpal, Ulceration, Fistula, Abscess, index.

ALA: Activity Lesion Assessment.

[38]. Finally, the Passos-Soares study reported that periodontal disease combined with tooth decay was significantly associated with increased involvement of OHRQoL.

### Temporomandibular dysfunction and OHRQoL

Only one study [41] evaluated the association between temporomandibular dysfunction and OHRQoL (Table 1) measured with the OHIP-14 instrument in Brazil in people from 16 to 80 years of age. Three types of temporomandibular dysfunction were recorded: muscle disorder, jaw displacement and arthralgia, and arthritis and osteoarthritis. The study reported that the presence of muscle disorder had a significant negative impact on OHRQoL, while the presence of mandibular displacement and arthralgia, and arthritis and osteoarthritis had no significant impact on OHRQoL.

### Salivary gland pathology and OHRQoL

Only one study reported the impact of salivary gland pathologies (presence of xerostomy) on OHRQoL in Chile (Table 1) in individuals aged 18 to 63. The total OHIP-14 score was used for OHRQoL measurement. The study reported that the presence of xerostomy significantly reduces OHRQoL compared to the absence of this disease [53].

### Cleft lip, cleft palate and OHRQoL

Only one case control study that assessed the association between cleft lip, cleft palate and OHRQoL was identified (Table 1) [40]. The study was carried out in Brazil in children between 8 and 10 years old. The cases were represented by children with cleft lip and palate belonging to the Hospital of Rehabilitation of Craniofacial anomalies and the University of Sao Paulo, while the controls were children without this condition recruited from national schools. The association between orofacial dysfunction and OHRQoL was evaluated in children with cleft lip and palate. For the measurement of orofacial dysfunction, the NOT-S instrument was used, based on the dimensions of sensory function, breathing, habits, chewing and swallowing, drooling and dry mouth, and CPQ 8–10 was used for OHRQoL. Significant differences were only identified in social welfare areas, but not in the overall outcome of CPQ 8–10, among children with and without orofacial dysfunction.

### Edentulism and OHRQoL

Two studies conducted in Brazil evaluated the association between edentulism and OHRQoL (Table 1). One study [46] included a population of individuals over the age of 12 and reported that a loss of previous teeth was negatively associated with OHRQoL. The second study evaluated included adults between the age of 20 and 64 and reported no significant association between the number of teeth lost and OHRQoL [21].

### Quality assessment of studies

With regard to the evaluation of the quality of the studies, eight were of good quality [20, 21, 26, 31–33, 49, 53], 22 were of intermediate quality [16–19, 22, 23, 25, 27–29, 34, 35, 37, 42, 43, 45, 47, 48, 50, 51, 54, 55] and ten were of poor quality [24, 30, 34, 36, 38, 39, 41, 44, 46, 52] (Table 3).

## Discussion

The purpose of this study was to conduct a systematic review to assess the impact of oral diseases on OHRQoL in people of countries from LAC. Forty studies with an observational

**Table 3. Quality assessment of the studies included.**

| N | Year | Author | Study Design | Q1 | Q2 | Q3 | Q4 | Q5 | Q6 | Q7 | Q8 | Q9 | Q10 | Q11 | Q12 | Q13 | Q14 | Rating |
|---|---|---|---|---|---|---|---|---|---|---|---|---|---|---|---|---|---|---|
| 1 | 2013 | Alves et al. | Cross-sectional | Y | Y | NR | Y | Y | N | N | Y | Y | NA | Y | NA | NA | Y | Good |
| 2 | 2014 | Tomazoni et al. | Cross-sectional | Y | N | NR | Y | Y | N | N | NA | Y | N | Y | NA | NA | Y | Fair |
| 3 | 2016 | Da Rosa et al. | Cross-sectional | Y | Y | Y | Y | Y | N | N | Y | Y | NA | Y | NA | NA | Y | Good |
| 4 | 2014 | López Ramos & García Rupaya. | Cross-sectional | Y | N | NR | Y | N | N | N | Y | Y | NA | Y | NA | NA | N | Poor |
| 5 | 2018 | Piva et al. | Cohort** | Y | Y | Y | Y | Y | Y | Y | Y | Y | N | Y | NA | N | N | Fair |
| 6 | 2017 | Feldens et al. | Cross-sectional | Y | N | Y | Y | Y | N | N | Y | Y | NA | Y | NA | NA | Y | Fair |
| 7 | 2017 | Martins et al. | Case-control* | Y | N | NR | Y | Y | Y | Y | Y | N | Y | N | NA | — | — | Fair |
| 8 | 2016 | Feldens et al. | Cross-sectional | Y | Y | NR | Y | N | N | N | Y | Y | NA | Y | NA | NA | Y | Fair |
| 9 | 2016 | Oliveira de Lima et al. | Cross-sectional | Y | Y | Y | Y | N | N | N | N | Y | NA | Y | NA | NA | N | Poor |
| 10 | 2013 | Kramer et al. | Cross-sectional | Y | Y | Y | Y | Y | N | N | Y | Y | NA | Y | NA | NA | Y | Good |
| 11 | 2014 | Abanto et al. | Cross-sectional | Y | N | Y | Y | Y | N | N | Y | Y | NA | Y | NA | NA | Y | Fair |
| 12 | 2013 | Martins-Júnior et al. | Cross-sectional | Y | N | Y | Y | Y | N | N | Y | Y | NA | Y | NA | NA | Y | Fair |
| 13 | 2017 | Simões et al. | Cross-sectional | Y | Y | Y | Y | Y | N | N | Y | Y | NA | Y | NA | NA | Y | Good |
| 14 | 2018 | Dutra et al. | Cross-sectional | Y | N | Y | Y | Y | N | N | Y | Y | NA | Y | NA | NA | N | Fair |
| 15 | 2011 | Abanto et al. | Cross-sectional | Y | N | Y | Y | N | N | N | Y | Y | NA | Y | NA | NA | Y | Fair |
| 16 | 2016 | Firmino et al. | Case-control* | Y | N | Y | Y | Y | Y | Y | NA | Y | Y | N | NA | ———— | ———— | Poor |
| 17 | 2015 | Meusel et al. | Cross-sectional | Y | Y | Y | Y | NR | N | N | N | N | NA | Y | NA | NA | N | Poor |
| 18 | 2016 | Rocha et al. | Cross-sectional | Y | Y | Y | NR | N | N | N | N | N | N | Y | NA | NA | N | Poor |
| 19 | 2014 | Batista et al. | Cross-sectional | Y | Y | Y | Y | Y | N | N | Y | Y | NA | Y | NA | NA | Y | Good |
| 20 | 2015 | Abanto et al. | Cross-sectional | Y | N | Y | Y | Y | N | N | Y | Y | NA | Y | NA | NA | Y | Fair |
| 21 | 2011 | Aldrigui et al. | Cross-sectional | Y | N | Y | Y | N | N | N | Y | Y | NA | Y | NA | NA | Y | Fair |
| 22 | 2019 | Milani et al. | Cross-sectional | Y | N | NR | Y | N | N | N | NA | Y | NA | Y | NA | NA | N | Poor |
| 23 | 2015 | Vieira-Andrade et al. | Case-control* | Y | N | CD | Y | Y | NR | Y | Y | N | Y | N | Y | ———— | ———— | Fair |
| 24 | 2014 | Ramos-Jorge et al. | Cross-sectional | Y | N | Y | Y | Y | N | N | N | N | NA | Y | NA | NA | Y | Poor |
| 25 | 2014 | Ramos-Jorge et al. | Cross-sectional | Y | N | Y | N | Y | N | N | Y | Y | NA | Y | NA | NA | Y | Fair |
| 26 | 2018 | Corrêa-Faria et al. | Cross-sectional | Y | N | Y | Y | Y | N | N | Y | Y | N | Y | NA | NA | Y | Fair |
| 27 | 2016 | Guedes et al. | Cohort** | Y | Y | Y | Y | Y | Y | Y | Y | Y | NA | Y | NA | N | Y | Good |
| 28 | 2013 | Sardenberg et al. | Cross-sectional | Y | Y | Y | Y | Y | N | N | N | Y | NA | Y | NA | NA | Y | Fair |
| 29 | 2013 | Scapini et al. | Cross-sectional | Y | N | Y | Y | N | N | N | Y | Y | NA | Y | NA | NA | N | Fair |
| 30 | 2017 | Bittencourt et al. | Cross-sectional | Y | N | Y | Y | Y | N | N | Y | Y | NA | Y | NA | NA | Y | Fair |
| 31 | 2016 | Naidu et al. | Cross-sectional | Y | N | Y | Y | Y | N | N | Y | N | NA | Y | NA | NA | Y | Fair |
| 32 | 2014 | Bendo et al. | Case-control* | Y | N | Y | NR | Y | Y | N | Y | Y | Y | N | Y | ———— | ——— | Fair |
| 33 | 2015 | Freire-Maia et al. | Cross-sectional | Y | Y | Y | Y | Y | N | N | Y | Y | NA | Y | NA | NA | Y | Good |
| 34 | 2019 | Montes et al. | Case-control* | Y | Y | Y | Y | Y | Y | NR | Y | Y | Y | NA | N | ———— | ——— | Fair |
| 35 | 2018 | Díaz et al. | Cross-sectional | Y | N | Y | Y | Y | N | N | Y | Y | NA | Y | NA | NA | Y | Fair |
| 36 | 2019 | Maia et al. | Cross-sectional | Y | N | N | Y | N | N | N | Y | Y | NA | Y | NA | NA | N | Poor |
| 37 | 2017 | Niklander et al. | Case-control* | Y | Y | Y | Y | Y | Y | CD | Y | Y | Y | N | Y | ———— | ———— | Good |
| 38 | 2019 | Bretz et al. | Cross-sectional | Y | N | Y | Y | N | N | N | Y | Y | NA | Y | NA | NA | N | Poor |
| 39 | 2018 | Passos-Soares et al. | Cross-sectional | Y | Y | NR | Y | Y | N | N | N | Y | NA | Y | NA | NA | Y | Fair |
| 40 | 2012 | Martins-Júnior et al. | Cross-sectional | Y | N | NR | Y | Y | N | N | N | Y | NA | Y | NA | NA | N | Poor |

* Quality Assessment Tool for Case-control Studies.

** Quality Assessment Tool for Observational Cohort and Cross-Sectional Studies.

Y: Yes; N: No; CD: Cannot determine; NA: Not applicable; NR: Not reported.

Q: Question of the study assessment tools.

design were included, 36 of which were developed in Brazil, one in Peru, one in Chile, one in Trinidad and Tobago, and one in Colombia. Of the total studies included, 30 evaluated tooth decay, 20 malocclusion, 22 TDI, three periodontal diseases, one cleft lip and palate, one tempo-romandibular dysfunction, two edentulism, and one salivary gland pathology. Twenty-five studies included children, nine were performed in adolescents, five were on adults, and one study evaluated more than one age group. The OHRQoL instruments most commonly used in the studies evaluated were CPQ11-14, ECOHIS, and B-ECOHIS. Despite the variety of instruments used to measure OHRQoL, and the different oral disease definitions of the studies included, most of the studies evaluated reported that OHRQoL was affected by the different oral diseases evaluated in children, adolescents and adults in concordance with the results of previous reviews with information specific to the LAC [57, 58].

Some negative associations between the presence of tooth decay and OHRQoL [16–19, 21, 23, 25, 27, 28, 30–37, 42, 43, 45, 47–50, 52, 54, 55] were identified in 27 of 30 studies included. These results have been reported in previous systematic reviews evaluating pre-school children and adolescents, in which this association was also considered as negative [7, 59]. This finding is of interest in LAC, as it is one of the regions worldwide with the highest prevalence and incidences of untreated tooth decay in deciduous and permanent teeth [3]. In addition, within the same region the prevalence of cavities in deciduous and permanent teeth is unequally distributed, with the prevalence being 10% to 20% lower in Brazil compared to other countries in LAC [60]. Thus, there is a need for a greater number of studies on the relationship between tooth decay and OHRQoL in other countries in the region, since 90% of the studies evaluated in this review were conducted in Brazil, and the problem related to cavities seems to be of greater magnitude in other countries of the region.

Moreover, only seven of the 20 studies included evaluation of the relationship between malocclusion and OHRQoL and concluded that the greater the degree of severity or the presence of malocclusion the greater the impact on OHRQoL [23, 24, 26, 27, 47–49]. However, most studies that did not identify this association did take into account the analysis of the presence or absence of disease; that is, individuals identified with the condition may have had both mild and severe forms, which may have led to an underestimation of the associations identified, bearing in mind that two of the studies evaluated described a relationship between malocclusion and OHRQoL according to the severity of malocclusion and reported that only the most severe forms negatively impacted OHRQoL [24, 49]. On the other hand, malocclusion is commonly associated with cavities and periodontal disease, mainly because tooth bad positions promote bacterial plaque buildup and hinder proper oral hygiene [61] and thus, OHRQoL can be even more affected by the presence of other comorbidities. In addition, some parents of children or adolescents with malocclusion may underestimate the impact of this disease, and it may remain unnoticed except in the case of obvious abnormalities which have a psychological or social impact on the sufferer [62] while less severe cases, remain unnoticed. Also, it is relevant that LAC is mainly composed of low- and middle-income countries, which is an important factor considering the high cost of orthodontic treatments and maxillary orthopedics, which are generally not covered by the public health systems of this region, thereby reducing the probability of these patients receiving treatment.

With regard to the relationship between TDI and OHRQoL, of the 22 studies evaluated, 13 identified a significant negative association between TDI and OHRQoL [17–19, 22, 29–31, 33, 34, 39, 44, 45, 55] while nine studies did not [16, 23, 28, 35, 48–52]. Most of these latter studies did not include adolescents but rather school- and preschool-age children, and therefore, the impact of TDI may not have been significant given that this age group may have less aesthetic concerns and less social impact in the presence of fractures or previous tooth loss as a result of a TDI. As in our study, previous systematic reviews have also shown heterogeneous results in

regard to this association. While the systematic review by Borges et al. described a negative impact of this pathology on OHRQoL in preschool children [8], the systematic review by Antunes RS [63] reported that the negative association between TDI and OHRQoL was significant for all dimensions of OHRQoL only in pre-adolescents between the ages of 11 and 14, and when the CPQ 11–14 instrument is used, while only the symptom domain of the OHRQoL is affected in children. Therefore, the association between TDI and OHRQoL may vary depending on the age of the population and the OHRQoL measuring instruments used [8, 63]. Higher-level methodological studies and long-term follow-up are required to learn more about this association and the groups that are most affected.

In general, studies reporting on the impact of periodontal diseases (n-2) [38, 50], temporomandibular dysfunction (n-1) [41], salivary gland pathology (n-1) [53], cleft lip and palate(n-1) [40] and edentulism (n-2) [21, 46] on OHRQoL are scarce in the population of LAC. In this regard, in countries of other regions in the world, mainly developed countries, a greater number of studies have been reported in relation to the association of periodontal diseases, edentulism and xerostomy with OHRQoL [64]. However, in addition to the psychological and social repercussions of these oral diseases, they also may have consequences in the general health. Previous reviews have demonstrated this association. For example, the systematic review of Stepan et al. [65] reported that in addition to the psychological effect, temporomandibular dysfunction may cause pain, synovitis, limitation of the opening of the mouth and locking of the jaw. Likewise, Romandini et al. [66] reported that periodontal diseases and its more advanced sequela, edentulism, are associated with a higher risk of mortality from cardiovascular diseases, neoplasms, cerebrovascular diseases and respiratory infections. Also, in addition to the difficulty in chewing and speaking caused by cleft lip and palate, this condition is associated with recurrent ear infections, and even hearing loss that can generate a negative effect on the OHRQoL as reported by the systematic review of Al-Namankany and Alhubaishi [67]. On the other hand, Pina et al. [68] concluded that to date, because of the limited evidence available, it is not clear to what extent a reduction in the level of salivary production impairs the OHRQoL. This highlights the need to perform further studies on the relationship of the diseases mentioned and OHRQoL in the population of LAC, with the aim of understanding the extent to which these diseases and their consequences affect OHRQoL, and to allow developing informed measures regarding these conditions at the health system level.

It is important to mention that of the 42 countries that make up LAC, only 5 are represented in this study, with 90% of the studies having been conducted in Brazil. This highlights the presence of a significant research gap on the subject of OHRQoL in the region of LAC countries, which has previously been identified in the area of dental public health [69], as well as in relation to scientific production in general, with Brazil having greater representation [70]. This is probably due to investment in research in Brazil, which reaches approximately 1.3% of the GDP, with the largest percentage in the region and with approximately two-thirds of total full-time researchers in South America [70]. Therefore, an adaptation of this measures to other contexts in the region is important so as not to leave other countries behind. On the other hand, in previous systematic reviews assessing the impact of oral diseases on general quality of life and OHRQoL globally, the representativeness of LAC countries is very low [7–9, 59], except for Brazil, demonstrating the lack of research in the area of oral health in this region.

The most common instruments used to measure OHRQoL were CPQ11-14 (n-8), ECOHIS (n-8) and B-ECOHIS (n-8). In this regard, a systematic review carried out by Zaror *et al.* (2018) made a standardized comparison of the instruments available for evaluating OHRQoL in the population of children and adolescents, which included 18 instruments evaluated using the *Evaluating Measures of Patient-Reported Outcomes* (EMPRO) tool. The study concluded

that ECOHIS in preschool children and CPQ11-14 were the best rated with good reliability, responsiveness and interpretability [71]. Another reason for the frequent use of these instruments is that they have been validated in several languages, including English, Portuguese and Spanish, allowing them to be widely used in other regions, with the exception of B-ECOHIS which can only be applied in Brazil [72]. On the other hand, although Spanish as the first language is predominant in LAC, there is a wide variety of intraregional dialects, which requires translation, adaptation and validation procedures of the OHRQoL instruments available in that language, being a preliminary step necessary for the development of studies on this subject in the different countries of the region.

Among the main limitations of this systematic review are the differences in diagnostic criteria for oral diseases, the instruments used for OHRQoL evaluation, and measures of association reported in the studies included. Given this heterogeneity, the usefulness of combining association estimators through meta-analysis would be very limited. On the other hand, some studies published as grey literature or in thematic and regional bibliographic databases that were not considered in the search for this systematic review may not have been included. Nonetheless, this is one of the first systematic reviews to assess the impact of oral diseases on OHRQoL in the context of LAC, which, in similar previously published systematic reviews, has had little representativeness [7–9, 58, 64]. In addition, an extensive systematic search for information was conducted in five databases, including LILACS, which houses much of the peer-reviewed research in LAC, in order to achieve maximum representativeness of studies on the subject from countries in the region. Similarly, this study included the evaluation of oral conditions, and age groups, not considered in previous systematic reviews. Furthermore, the 10-year cut-off for the publication date of the eligible studies could be considered as a limitation. However, we took into account the fact that most of the studies validating OHRQoL instruments in LAC were published in the first decade of the 2000s [73–77]. Therefore, it is be expected that within only a few years after validation these instruments began to be used in larger-scale studies. Finally, since this systematic review represents an overview of studies evaluating the impact of various oral conditions on OHRQoL, we recommend that future studies should perform calculations of association estimators taking into account the impact of specific oral diseases on OHRQoL.

In conclusion, this systematic review shows that most studies in LAC identified a negative relationship between the presence and/or severity of oral conditions and OHRQoL. However, there is an under-representation in the scientific research in this area in most of the countries in the region. This problem requires the implementation of research, science and technology policies, together with increased investment at the state level, in order to encourage the development of these studies, which are of social interest and with a public health approach. In addition, a greater number of prospective longitudinal studies and including adult and older adult populations are needed, to further provide information to enable better decision-making and investment of the limited health budgets that are common in most LAC countries.

## Supporting information

**S1 Checklist.**
(DOC)

**S1 Table. Search Strategies.**
(DOCX)

**S1 File. Data file of the study.**
(TXT)

## Acknowledgments

The authors are grateful to Donna Pringle for reviewing the language and style.

## Author Contributions

**Conceptualization:** Diego Azañedo, Akram Hernández-Vásquez.

**Data curation:** Daniel Comandé.

**Formal analysis:** María T. Yactayo-Alburquerque, María L. Alen-Méndez, Diego Azañedo, Daniel Comandé, Akram Hernández-Vásquez.

**Investigation:** María T. Yactayo-Alburquerque, María L. Alen-Méndez, Diego Azañedo, Akram Hernández-Vásquez.

**Methodology:** Diego Azañedo, Daniel Comandé, Akram Hernández-Vásquez.

**Project administration:** Diego Azañedo, Akram Hernández-Vásquez.

**Software:** Akram Hernández-Vásquez.

**Supervision:** Diego Azañedo, Akram Hernández-Vásquez.

**Validation:** María T. Yactayo-Alburquerque, María L. Alen-Méndez, Diego Azañedo, Daniel Comandé, Akram Hernández-Vásquez.

**Visualization:** María T. Yactayo-Alburquerque, María L. Alen-Méndez, Diego Azañedo, Daniel Comandé, Akram Hernández-Vásquez.

**Writing – original draft:** María T. Yactayo-Alburquerque, María L. Alen-Méndez, Diego Azañedo, Daniel Comandé, Akram Hernández-Vásquez.

**Writing – review & editing:** María T. Yactayo-Alburquerque, María L. Alen-Méndez, Diego Azañedo, Daniel Comandé, Akram Hernández-Vásquez.

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
