## [Decision Letter · Decision Letter 0]

21 Apr 2021

PONE-D-21-08291

Impact of oral diseases on oral health-related quality of life: A systematic review of studies conducted in Latin America and the Caribbean

PLOS ONE

Dear Dr. Yactayo,

Thank you for submitting your manuscript to PLOS ONE. After careful consideration, we feel that it has merit but does not fully meet PLOS ONE’s publication criteria as it currently stands. Therefore, we invite you to submit a revised version of the manuscript that addresses the points raised during the review process.

We look forward to receiving your revised manuscript.

Kind regards,

Frédéric Denis, Ph.D.

Academic Editor

PLOS ONE

Journal Requirements:

Reviewers' comments:

Reviewer's Responses to Questions

**Comments to the Author**

1. Is the manuscript technically sound, and do the data support the conclusions?

Reviewer #1: Yes

Reviewer #2: Partly

2. Has the statistical analysis been performed appropriately and rigorously? 

Reviewer #1: Yes

Reviewer #2: N/A

3. Have the authors made all data underlying the findings in their manuscript fully available?

Reviewer #1: No

Reviewer #2: Yes

4. Is the manuscript presented in an intelligible fashion and written in standard English?

Reviewer #1: Yes

Reviewer #2: Yes

5. Review Comments to the Author

Reviewer #1: Nicely organized and conducted study. Below are the review necessary:

1) Materials and methods

-Date extraction process

Where it reads “in the Microsoft Excel” add between parentheses: Microsoft Corp.; https://products.office.com;

-The authors should review the text and change the type of study to design of study;

-The authors should analyze the references of the papers selected to find other references. This action is important to verify other references, which can be out of sought during the execution of search strategy. Moreover, the authors should carry out the description in material and methods;

-Evaluation quality of study: the authors should describe clearer the criteria used to evaluation;

2- Discussion

-This sentence: “Given this heterogeneity, the usefulness of combining association estimators through meta-analysis would be very limited”. It is a methodological feature. Therefore, the authors should transfer this sentence to materials and methods, and adapting in the text;

The authors should:

-include the gray literature in the material and method. In addition, the authors should perform the search in Google Scholar. Moreover, the authors need to realize a manual search of the references in the databases;

-include in the limitation of study: the work has done with the papers published over the past 10 years. What are the reasons for this?

Reviewer #2: This is a beautiful piece of work, coming from a leader in this field that conduct a systematic review to assess the impact of oral diseases on OHRQoL.This systematic review is not only for children, but also for adults and the order adults.

In addition to traditional tooth decay TDI, and malocclusion, this study also evaluated periodontal disease, temporomandibular dysfunctions, pathologies of the salivary glands, cleft lip and palate, and edentulism.

However, please revise the following points.

It would be better to indicate the publication language and status of the adopted papers and their data extraction processes.

Was there any method to assess the risk of bias that might affect the evidence for an individual study?

Isn't it necessary to show more details other than the OHRQoL measuring?

Is it possible to provide efficacy estimates and confidence intervals for each intervention group for each study for all outcomes examined for study results?

The number of papers included in the systematic review on the presence or absence of periodontal disease, temporomandibular dysfunction, pathologies of the salivary glands, cleft lip and palate and edentulism was too small. Regarding these papers, the authors should discuss not only from a social perspective but also from a medical perspective.

Please reconsider the number of participants and the presentation period for the Eligibility criteria. After the reconsideration, please include more literature in the review.

Wouldn't it be necessary to describe the systematic review funder, other support such as data provision, and the role of the funder in the systematic review?

It would be very exciting if each oral disease actually had a more significant effect on OHRQoL.

6. PLOS authors have the option to publish the peer review history of their article (what does this mean?). If published, this will include your full peer review and any attached files.

Reviewer #1: No

Reviewer #2: No

---

## [Author Response · Author response to Decision Letter 0]

15 May 2021

May 15, 2021

Frédéric Denis, Ph.D.

Academic Editor

PLOS ONE

Ref: Submission [PONE-D-21-08291] - [EMID:4b3237bc98cc04df]

Title: “Impact of oral diseases on oral health-related quality of life: A systematic review of studies conducted in Latin America and the Caribbean”.

Dear Dr. Denis;

We thank the reviewers and the Editor-in-Chief for their helpful comments and suggestions regarding our manuscript, which have all been addressed. Therefore, we are submitting a revised version of the manuscript. The point-by-point responses and changes are shown below. 

Answer: Thank you for the suggestion. After a full review of the manuscript and the tables and bibliographic references, we confirm that it meets the PLOS ONE style requirements.

Reviewer #1: Nicely organized and conducted study. Below are the review necessary:

1) Materials and methods

Question 1: Date extraction process: Where it reads “in the Microsoft Excel” add between parentheses: Microsoft Corp.; https://products.office.com;

Answer: Thank you for your recommendation. We have added the reference directly in the manuscript. Now it reads: in the Microsoft Excel program (Microsoft Corporation, Redmond, Washington, USA).

Question 2: The authors should review the text and change the type of study to design of study;

Answer: Thank you for making this known to us. We have changed the phrase “type of study” to “study design” throughout the manuscript.

Question 3: The authors should analyze the references of the papers selected to find other references. This action is important to verify other references, which can be out of sought during the execution of search strategy. Moreover, the authors should carry out the description in material and methods;

Answer: Thank you for bringing this to our attention. A review was made of the list of references from the papers included in the systematic review. Actually, this was stated in our study protocol published in PROPERO (CRD42020156098). We apologize for not having mentioned it in the manuscript.

We have added the following sentence in the “Search strategies” subsection of the “Materials and methods” section: “An additional search in the reference lists of the studies included was carried out to identify other publications not identified by the systematic search”

Question 4: Evaluation quality of study: the authors should describe clearer the criteria used to evaluation;

Answer: Thank you for this observation. In the manuscript, we have further explained the criteria used to evaluate the quality of the included studies. The related changes can be found in the “evaluation of study quality” subsection from the “Materials and methods” section. Now it reads:

“Two authors (MTYA and MLAM) independently evaluated the quality of the studies using the U.S. National Institutes of Health tool [14]. This tool presents a different number of questions depending on the design of the study evaluated (14 questions for cohort or cross-sectional studies, and 12 for case-control studies). Each question has five possible answers: 'yes', 'no', 'cannot be determined', 'not applicable' and 'not determined'. After completing the questions, each article was graded as good, fair, or poor quality. Disagreements were resolved by consensus with a third reviewer (DA).”

2) Discussion

Question 5: This sentence: “Given this heterogeneity, the usefulness of combining association estimators through meta-analysis would be very limited”. It is a methodological feature. Therefore, the authors should transfer this sentence to materials and methods, and adapting in the text;

Answer: Thank you for this observation. We have added the solicited information in the “Synthesis of results” subsection in the “Materials and methods” section. Now it reads: “Due to the heterogeneity among the studies included, we considered a meta-analysis inappropriate, and therefore, we focused on the qualitative synthesis of the studies”

Question 6: The authors should: include the gray literature in the material and method. In addition, the authors should perform the search in Google Scholar. Moreover, the authors need to realize a manual search of the references in the databases;

Answer: Thank you for this recommendation. While the Cochrane handbook for Systematic Reviews specifies that a gray literature search is desirable, this is a recommendation for intervention reviews https://training.cochrane.org/handbook/current/chapter-04#section-4-3. This is not the case of this study, which deals in its entirety with observational studies. Furthermore, even in systematic reviews of interventions, the inclusion of gray literature according to the mentioned Cochrane handbook is not mandatory, but rather desirable. Therefore, we decided not to include gray literature in our systematic review. This decision was made taking into account that most gray literature is not peer reviewed, and neither is it indexed in major bibliographic sources, which can greatly undermine the quality of the studies in this category. 

With regard to the manual search of the references of the studies included, as stated in our protocol, this was done while conducting this systematic review. This information has been added to the “Selection of studies” subsection, “Materials and Methods” section of the manuscript. The added information reads “An additional search in the reference lists of the studies included was carried out to identify other publications not identified by the systematic search.”

Question 7: The authors should include in the limitation of study: the work has been done with the papers published over the past 10 years. What are the reasons for this?

Answer: Thank you for the recommendation. In this case we chose this period of time because most of the OHRQoL instruments were validated in the Latin American and the Caribbean region around the first half decade of the 2000s. Below you can find some references supporting this:

1. Dini EL, McGrath C, Bedi R. An evaluation of the oral health quality of life (OHQoL) instrument in a Brazilian population. Community Dent Health. 2003 Mar;20(1):40-4. PMID: 12688603.

2. Souza RF, Patrocínio L, Pero AC, Marra J, Compagnoni MA. Reliability and validation of a Brazilian version of the Oral Health Impact Profile for assessing edentulous subjects. J Oral Rehabil. 2007 Nov;34(11):821-6. doi: 10.1111/j.1365-2842.2007.01749.x. PMID: 17919248.

3. Goursand D, Paiva SM, Zarzar PM, Ramos-Jorge ML, Cornacchia GM, Pordeus IA, Allison PJ. Cross-cultural adaptation of the Child Perceptions Questionnaire 11-14 (CPQ11-14) for the Brazilian Portuguese language. Health Qual Life Outcomes. 2008 Jan 14;6:2. doi: 10.1186/1477-7525-6-2. PMID: 18194552; PMCID: PMC2246108.

4. Oliveira BH, Nadanovsky P. Psychometric properties of the Brazilian version of the Oral Health Impact Profile-short form. Community Dent Oral Epidemiol. 2005 Aug;33(4):307-14. doi: 10.1111/j.1600-0528.2005.00225.x. PMID: 16008638.

5. Lopez R, Baelum V. Spanish version of the Oral Health Impact Profile (OHIP-Sp). BMC Oral Health. 2006 Jul 7;6:11. doi: 10.1186/1472-6831-6-11. PMID: 16827940; PMCID: PMC1534011.

6. Pires CP, Ferraz MB, de Abreu MH. Translation into Brazilian Portuguese, cultural adaptation and validation of the oral health impact profile (OHIP-49). Braz Oral Res. 2006 Jul-Sep;20(3):263-8. doi: 10.1590/s1806-83242006000300015. PMID: 17119711.

Therefore, it is to be expected that within only a few years after validation these instruments began to be used in larger-scale studies. Likewise, even taking into account the 10-year cut-off as the inclusion criterion for the studies evaluated, 40 studies were included in our systematic review, which is a considerably large number. However, we have mentioned that the use of this cut-off may be a limitation in our systematic review. We added the following sentence to the limitations paragraph of the Discussion section of the manuscript:

“Furthermore, the 10-year cut-off for the publication date of the eligible studies could be considered as a limitation. However, we took into account the fact that most of the studies validating OHRQoL instruments in LAC were published in the first half decade of the 2000s. Therefore, it is to be expected that within only a few years after validation these instruments began to be used in larger-scale studies.”

Reviewer #2: 

This is a beautiful piece of work, coming from a leader in this field that conduct a systematic review to assess the impact of oral diseases on OHRQoL.This systematic review is not only for children, but also for adults and the order adults.

In addition to traditional tooth decay TDI, and malocclusion, this study also evaluated periodontal disease, temporomandibular dysfunctions, pathologies of the salivary glands, cleft lip and palate, and edentulism.

However, please revise the following points:

Question 1: It would be better to indicate the publication language and status of the adopted papers and their data extraction processes.

Answer: Thank you for making this known to us. In the country column of Table 1 we have added the publication language. In relation to the status and data extraction processes, we must clarify that all the studies included correspond to completed and published studies, and thus, all the data extraction process was completed. For this reason, we did not consider adding this information to the manuscript.

Question 2: Was there any method to assess the risk of bias that might affect the evidence for an individual study?

Answer: Thank you for this question. We used the term “quality of the studies” and not “risk of bias” considering that most of the studies were observational and the tool used is precisely called “Study Quality Assessment Tool” and not “risk of bias” like other instruments. In addition, the MOOSE guidelines for Systematic Review of observational studies recommends reporting the “assessment of the quality of study”; therefore, as we have taken this last as a reference for the reporting of our systematic review, the term “risk of bias” was not considered.

Question 3: Isn't it necessary to show more details other than the OHRQoL measuring?

Answer: Thank you for the observation. In addition to the OHRQoL result, other variables such as: type of study, oral diseases, OHRQoL measurement instrument, age group and nationality were taken into consideration. However, the aim of this systematic review was to assess the impact of oral diseases on OHRQoL, and therefore, the report was focused on the last variable, because it was the outcome of interest in our study.

Question 4: Is it possible to provide efficacy estimates and confidence intervals for each intervention group for each study for all outcomes examined for study results?

Answer: Thank you for the commentary. Since no clinical trials have been included, it is not possible to calculate efficacy estimators. On the other hand, the main objective of this study was to present an overview of studies conducted in Latin America and the Caribbean that evaluated the impact of the largest possible number of oral diseases on OHRQoL. Therefore, a more detailed evaluation, performing the calculation of association estimators and taking into account specific oral diseases, should be addressed in future studies. We have added the following at the end of the limitation paragraph in the Discussion section:

“Since this systematic review represents an overview of studies evaluating the impact of various oral conditions on OHRQoL, we recommend that future studies should perform calculations of association estimators taking into account the impact of specific oral diseases on OHRQoL.”

Question 5: The number of papers included in the systematic review on the presence or absence of periodontal disease, temporomandibular dysfunction, pathologies of the salivary glands, cleft lip and palate and edentulism was too small. Regarding these papers, the authors should discuss not only from a social perspective but also from a medical perspective.

Answer: Thank you for the recommendation. In response, we have added the following information to the paragraph in which the studies about these conditions were mentioned:

“However, in addition to the psychological and social repercussions of these oral diseases, these also may have consequences in the general health. Previous reviews have demonstrated this association. For example, the systematic review of Stepan et al.[65] reported that, in addition to the psychological effect, temporomandibular dysfunction may cause pain, synovitis, limitation of the opening of the mouth and locking of the jaw. Likewise, Romandini et al. [66] reported that periodontal diseases and its more advanced sequela, edentulism, are associated with a higher risk of mortality from cardiovascular diseases, neoplasms, cerebrovascular diseases and respiratory infections. Also, in addition to the difficulty in chewing and speaking caused by cleft lip and palate, this condition is associated with recurrent ear infections, and even hearing loss that could generate a negative effect on the OHRQoL as reported by the systematic review of Al-Namankany and Alhubaishi.[67] On the other hand, Pina et al. [68] concluded that, to date, because of the limited evidence available, it is not clear to what extent a reduction in the level of salivary production impairs the OHRQoL. This highlights the need to perform further studies on the relationship of the diseases mentioned and OHRQoL in the LAC population, with the aim of understanding the extent to which these diseases and their consequences affect OHRQoL, and to allow developing informed measures regarding these conditions at the health system level.”

Question 6: Please reconsider the number of participants and the presentation period for the Eligibility criteria. After the reconsideration, please include more literature in the review.

Answer: An agreement was reached among the authors after reviewing other systematic reviews of prevalence using the 100 participants cut-off point or similar to select studies. Also, it must be taken into account that in studies with a low number of participants there may be a selection bias. Below there are some antecedents of studies using this type of cut-off for the number of participants required for inclusion of a study:

1. Captieux M, Prigge R, Wild S, Guthrie B (2020) Defining remission of type 2 diabetes in research studies: A systematic scoping review. PLoS Med 17(10): e1003396. https://doi.org/10.1371/journal.pmed.1003396.

2. Schneider, M., Molnar, A., Angeli, O., Szabo, D., Bernath, F., Hajdu, D., Gombocz, E., Mate, B., Jiling, B., Nagy, B.V., Nagy, Z.Z., Peto, T. and Papp, A. (2021), Prevalence of Cilioretinal Arteries: A systematic review and a prospective cross‐sectional observational study. Acta Ophthalmol, 99: e310-e318. https://doi.org/10.1111/aos.14592. 

3. J. Walker, C. Holm Hansen, P. Martin, A. Sawhney, P. Thekkumpurath, C. Beale, S. Symeonides, L. Wall, G. Murray, M. Sharpe. Prevalence of depression in adults with cancer: a systematic review. Annals of oncology. 2013 April; 24(4): 895-900. https://doi.org/10.1093/annonc/mds575.

Regarding the eligibility criteria considering the studies published in the last 10 years, we chose this cut-off because most of the OHRQoL instruments were validated in the Latin American and the Caribbean region in the first half decade of the 2000s. Below you can find some references supporting this affirmation:

7. Dini EL, McGrath C, Bedi R. An evaluation of the oral health quality of life (OHQoL) instrument in a Brazilian population. Community Dent Health. 2003 Mar;20(1):40-4. PMID: 12688603.

8. Souza RF, Patrocínio L, Pero AC, Marra J, Compagnoni MA. Reliability and validation of a Brazilian version of the Oral Health Impact Profile for assessing edentulous subjects. J Oral Rehabil. 2007 Nov;34(11):821-6. doi: 10.1111/j.1365-2842.2007.01749.x. PMID: 17919248.

9. Goursand D, Paiva SM, Zarzar PM, Ramos-Jorge ML, Cornacchia GM, Pordeus IA, Allison PJ. Cross-cultural adaptation of the Child Perceptions Questionnaire 11-14 (CPQ11-14) for the Brazilian Portuguese language. Health Qual Life Outcomes. 2008 Jan 14;6:2. doi: 10.1186/1477-7525-6-2. PMID: 18194552; PMCID: PMC2246108.

10. Oliveira BH, Nadanovsky P. Psychometric properties of the Brazilian version of the Oral Health Impact Profile-short form. Community Dent Oral Epidemiol. 2005 Aug;33(4):307-14. doi: 10.1111/j.1600-0528.2005.00225.x. PMID: 16008638.

11. Lopez R, Baelum V. Spanish version of the Oral Health Impact Profile (OHIP-Sp). BMC Oral Health. 2006 Jul 7;6:11. doi: 10.1186/1472-6831-6-11. PMID: 16827940; PMCID: PMC1534011.

12. Pires CP, Ferraz MB, de Abreu MH. Translation into Brazilian Portuguese, cultural adaptation and validation of the oral health impact profile (OHIP-49). Braz Oral Res. 2006 Jul-Sep;20(3):263-8. doi: 10.1590/s1806-83242006000300015. PMID: 17119711.

Therefore, it is to be expected that within only a few years after validation these instruments began to be used in larger-scale studies. Nonetheless, even considering this cut-off, the number of studies included in our systematic review was considerable (40 studies).

Question 7: Wouldn't it be necessary to describe the systematic review funder, other support such as data provision, and the role of the funder in the systematic review?

Answer: Thank you for the question. This was not included, because this was a self-funded systematic review. For the process of accessing and searching the databases, we had the support of a librarian (DC).

Question 8: It would be very exciting if each oral disease actually had a more significant effect on OHRQoL.

Answer: According to our findings, evidence does support an association between different oral diseases, such as dental decay, malocclusion, and TDI, and OHRQoL. However, most of the studies included were cross-sectional, and thus, causality could not be established. Therefore, it is necessary to develop prospective studies to establish a relationship between oral diseases and OHRQoL over time.

The Authors.

---

## [Decision Letter · Decision Letter 1]

19 May 2021

Impact of oral diseases on oral health-related quality of life: A systematic review of studies conducted in Latin America and the Caribbean

PONE-D-21-08291R1

Dear Dr. Yactayo,

We’re pleased to inform you that your manuscript has been judged scientifically suitable for publication and will be formally accepted for publication once it meets all outstanding technical requirements.

Kind regards,

Frédéric Denis, Ph.D.

Academic Editor

PLOS ONE

Additional Editor Comments (optional):

Reviewers' comments:

Reviewer's Responses to Questions

**Comments to the Author**

1. If the authors have adequately addressed your comments raised in a previous round of review and you feel that this manuscript is now acceptable for publication, you may indicate that here to bypass the “Comments to the Author” section, enter your conflict of interest statement in the “Confidential to Editor” section, and submit your "Accept" recommendation.

Reviewer #1: All comments have been addressed

Reviewer #2: All comments have been addressed

2. Is the manuscript technically sound, and do the data support the conclusions?

Reviewer #1: Yes

Reviewer #2: Yes

3. Has the statistical analysis been performed appropriately and rigorously? 

Reviewer #1: N/A

Reviewer #2: Yes

4. Have the authors made all data underlying the findings in their manuscript fully available?

Reviewer #1: Yes

Reviewer #2: Yes

5. Is the manuscript presented in an intelligible fashion and written in standard English?

Reviewer #1: Yes

Reviewer #2: Yes

6. Review Comments to the Author

Reviewer #1: (No Response)

Reviewer #2: (No Response)

7. PLOS authors have the option to publish the peer review history of their article (what does this mean?). If published, this will include your full peer review and any attached files.

Reviewer #1: No

Reviewer #2: No

---

## [Editor Report · Acceptance letter]

21 May 2021

PONE-D-21-08291R1 

Impact of oral diseases on oral health-related quality of life: A systematic review of studies conducted in Latin America and the Caribbean 

Dear Dr. Yactayo-Alburquerque:

I'm pleased to inform you that your manuscript has been deemed suitable for publication in PLOS ONE. Congratulations! Your manuscript is now with our production department. 

Kind regards, 

on behalf of

Dr. Frédéric Denis 

Academic Editor

PLOS ONE